# Safety and Efficacy Outcomes of Robotic, Laparoscopic, and Laparotomic Surgery in Severe Obese Endometrial Cancer Patients: A Network Meta-Analysis

**DOI:** 10.3390/cancers17122018

**Published:** 2025-06-17

**Authors:** Carlo Ronsini, Mario Fordellone, Eleonora Braca, Mariano Catello Di Donna, Maria Cristina Solazzo, Giuseppe Cucinella, Cono Scaffa, Pasquale De Franciscis, Vito Chiantera

**Affiliations:** 1Unit of Gynecologic Oncology, National Cancer Institute, IRCCS, Fondazione “G. Pascale”, 80131 Naples, Italy; mariano.didonna@istitutotumori.na.it (M.C.D.D.); giuseppe.cucinella@istitutotumori.na.it (G.C.); c.scaffa@istitutotumori.na.it (C.S.); vito.chiantera@istitutotumori.na.it (V.C.); 2Medical Statistics Unit, Department of Mental and Physical Health and Preventive Medicine, University of Campania “Luigi Vanvitelli”, 80138 Naples, Italy; mario.fordellone@unicampania.it; 3Department of Woman, Child and General and Specialized Surgery, University of Campania “Luigi Vanvitelli”, 80138 Naples, Italy; eleonora.braca@unicampania.it (E.B.); mariacristina.solazzo@studenti.unicampania.it (M.C.S.); pasquale.defranciscis@unicampania.it (P.D.F.)

**Keywords:** endometrial cancer, obesity, surgical approaches, mini-invasive surgery, post-operative complications

## Abstract

Obesity is one of the most significant challenges in the management of endometrial carcinoma, as it complicates surgical treatment by limiting technical options and increasing the risk of complications. Our analysis set out to determine which surgical approach is safest and most effective for patients with a BMI ≥ 40. While our meta-analysis did not identify a statistically significant difference in safety between the available surgical methods, minimally invasive surgery (MIS) demonstrated a favorable trend, with lower rates of intra-operative and post-operative complications, as well as a higher likelihood of achieving complete surgical staging with bilateral sentinel lymph node removal. Notably, our network meta-analysis—which allows an indirect comparison of all three approaches—found that robotic surgery appears to be the most effective in minimizing intra-operative complications and achieving complete lymph node staging while performing similarly to laparoscopy in reducing post-operative complications. These findings suggest that robotic surgery could be considered the gold standard in terms of safety and efficacy for this patient population, supporting the centralization of care in specialized centers equipped with robotic systems. However, it is important to recognize that access to robotic surgery may be limited by equipment availability and cost, potentially impacting equitable patient care.

## 1. Introduction

Grade 3 obesity, also called severe obesity, a body mass index (BMI) greater than or equal to 40 kg/m^2^, is a global and steadily increasing public health problem in both developed and developing countries [1,2]. This phenomenon is not without consequences, as obesity is strongly correlated with several diseases, including endometrial carcinoma [3]. In patients with this neoplasm, a high percentage of severe obesity is observed, often due to hyperestrogenism induced by adipose tissue, which can stimulate the growth of cancer cells in the endometrial lining [4].

Hysterectomy with lymph node staging is considered the gold standard for the management of patients with endometrial carcinoma, as it allows the tumor to be removed and the extent of the disease to be assessed through lymph node analysis [5,6,7]. In recent years, minimally invasive surgery, particularly laparoscopy and robotic surgery, has emerged as the gold standard following major European guidelines [5,6,7]. These techniques offer significant advantages, such as reduced post-operative pain, shorter hospital stays, and faster recovery [8].

However, minimally invasive surgery can present unique challenges in patients with severe obesity. Technical limitations associated with increased anatomical complexity and difficulties in visualization and pelvic organ access can complicate the surgery [9]. In addition, obese patients have increased inherent risks, such as anaesthesiological complications [10], infections [11], and ventilation issues [12], which may adversely affect surgical outcomes and post-operative quality of life. Therefore, it is crucial to carefully consider these factors when planning surgery in this high-risk population.

### Objective

Our study aims to conduct a systematic review of the literature and make a quantitative analysis of it using meta-analysis and network meta-analysis to decide which is the best surgical approach in the management of patients with severe obesity and endometrial carcinoma between mini-invasive robotic (Robot), laparoscopic (LPS), and laparotomic (LPT) approaches.

## 2. Material and Methods

The methods for this study were specified a priori based on the recommendations in the Preferred Reporting Items for Systematic Reviews and Meta-Analyses (PRISMA) extension statement for network meta-analyses (PRISMA-NMA) [13] and Cochrane Handbook for Systematic Reviews of Intervention [14]. We registered the review to the International Prospective Register of Systematic Reviews (PROSPERO) site for meta-analysis with protocol number CRD 395959.

### 2.1. Search Method

We performed a systematic search for articles about obesity and surgery in the Pubmed Database and Scopus Database in September 2024, using the following keywords and Medical Subject Heading (MeSH) terms (“Obesity” [Mesh] AND “Endometrial Neoplasms” [Mesh]) AND (“Laparoscopy” [Mesh] OR “Robotic Surgical Procedures” [Mesh]) without any date restriction. We made no restrictions on the country. We considered only English entirely published studies. Searches were also conducted on ClinicalTrials.gov, the Cochrane Central Register of Controlled Trials, and the World Health Organization’s International Clinical Trials Registry Platform (ICTRP) to include more randomized controlled trials. In addition, the gray literature was reviewed, including sources like NTIS and PsycEXTRA, to identify abstracts from both international and national conferences.

### 2.2. Study Selection

Study selection was made independently by authors MCS and EB. In case of discrepancy, CR decided on inclusion or exclusion. Inclusion criteria were (1) studies that included patients undergoing surgery with at least a hysterectomy for the treatment of endometrial cancer and a BMI ≥ 40; (2) studies that reported at least one outcome of interest (intra-operative complication; post-operative complication, severe complication; complete surgical staging); and (3) peer-reviewed articles published originally. We excluded non-original studies, preclinical trials, animal trials, abstract-only publications, and articles in languages other than English. If possible, the authors of studies that were only published as congress abstracts were tried to be contacted via email and asked to provide their data. We mentioned the studies selected and all reasons for exclusion in the Preferred Reporting Items for Systematic Reviews and Meta-Analyses (PRISMA) flowchart (Figure 1). We assessed all included studies regarding potential conflicts of interest.

### 2.3. Risk of Bias

The risk of bias in the included studies was assessed according to the criteria outlined in the Cochrane Handbook for Systematic Reviews of Interventions. Seven specific domains were examined, which were associated with potential bias in estimating treatment effects. The domains assessed were as follows: (1) random sequence generation, (2) allocation concealment, (3) blinding of participants and study personnel, (4) blinding of outcome assessors, (5) completeness of outcome data, (6) selective reporting, and (7) any other sources of bias. Studies were rated as having a “low risk,” “high risk,” or “unclear risk” of bias based on these criteria. The bias evaluation was conducted independently by three authors (M.F., C.R., and M.C.D.D.), with any disagreements resolved through discussion with a fourth reviewer (P.D.F.). A network meta-regression was performed to explore the potential influence of key study-level covariates, including mean BMI, menopausal status distribution, tumor stage distribution, and publication year. These covariates were included as potential effect modifiers in the analysis to reduce heterogeneity and improve the robustness of the treatment comparisons.

### 2.4. Primary and Secondary Endpoints

The primary endpoint of this study was to assess the rate of intra-operative complications in patients undergoing hysterectomy for endometrial carcinoma by laparotomic (LPT), laparoscopic (LPS), or robotic (Robot) approach. The intra-operative complication rate was calculated as the absolute number of unexpected adverse events that occurred during surgery out of the number of procedures performed. Secondary outcomes were the rate of post-operative complications for the three different methods, understood as the absolute number of complications occurring from the end of surgery to the first 30 post-operative days out of the number of patients undergoing surgery, The rate of severe complications, understood as complications ≥ grade 3 according to the Clavien–Dindo Classification for Surgical Complications [15]. The rate of complete surgical staging, defined as the bilateral removal of at least one sentinel lymph node if the surgical intent was sentinel lymph node retrieving, out of the total number of operations.

### 2.5. Statistical Analysis

Heterogeneity among the studies was tested using the chi-square test and I-square tests [16]. The odds ratio (OR) and 95% confidence intervals (CIs) were used for dichotomous variables. Random-effect models were applied to the type of studies included. Chi-square tests were used to compare continuous variables. Higgins I-squared (I^2^) index higher than 0% was used to target potential heterogeneity. In the cases of significant heterogeneity, sensitivity analyses were ruled out to understand the relevant sources of heterogeneity.

A network meta-analysis (NMA) was conducted to compare the relative effectiveness of all included interventions, integrating direct and indirect evidence from comparative studies. The analysis used a Bayesian framework, applying a random-effects model to account for potential heterogeneity across studies. Treatment effects were expressed as odds ratios (ORs) with 95% credible intervals (CIs) for dichotomous outcomes. Consistency between direct and indirect comparisons was assessed using node-splitting models, and global consistency was evaluated by comparing the fit of consistency and inconsistency models. The network geometry was visually inspected to ensure the connectivity of the evidence, and the transitivity assumption was assessed by comparing study and patient characteristics across treatment comparisons. Convergence of the Bayesian models was evaluated using trace plots and the Gelman–Rubin diagnostic, with a value of <1.05 indicating satisfactory convergence. We used the surface under the cumulative ranking curve (SUCRA) to rank the treatments, with higher values indicating a greater likelihood of being the most effective treatment. Statistical analyses were conducted using R software 4.3.3 (accessed on 2 February 2024) with the gemtc and netmeta packages, as well as WinBUGS for Bayesian computations.

### 2.6. Quality Assessment

We assessed the quality of the included studies using the Newcastle–Ottawa scale (NOS) [17]. This assessment scale uses three broad factors (selection, comparability, and exposure), with the scores ranging from 0 (lowest quality) to 8 (best quality). Two authors (C.R. and P.D.F.) independently rated the study’s quality. Any disagreement was subsequently resolved by discussion or consultation with LC. We reported the NOS Scale in Appendix A.

We used a funnel plot analysis to assess publication bias. We used Egger’s regression test [18] to determine the asymmetry of funnel plots (Appendix A).

### 2.7. Declaration of Generative AI in Scientific Writing

The authors declare that no AI was used to write the original draft. Grammar correction tools (Grammarly, Inc., San Francisco, CA, USA) were used to improve the quality of English and readability. The technology was used under human oversight and control.

## 3. Results

### 3.1. Studies’ Characteristics

After the database search, 309 articles matched the search criteria. After removing records with no full text, duplicates, and wrong study designs (e.g., reviews), 30 were eligible. Of those, 12 matched the inclusion criteria and were included in the systematic review. Four were non-comparative, single-armed studies, three about robotic approach, and one about LPS. The other eight were comparative studies between LPS and LPT or Robot hysterectomy for endometrial cancer and were included in quantitative analysis (Figure 1). One was comparing LPT vs. Robot, one was comparing LPS vs. robotic, three were comparing LPS vs. LPT, and three were comparing all three different approaches. The countries where the studies were conducted, the publication year range, the studies’ design, BMI ranges, type of surgical approach, FIGO staging, and number of participants are summarized in Table 1.

The quality of all studies was assessed by NOS (Appendix A). Overall, the publication years ranged from 2011 to 2021 and BMI from 40 to 75. Figo 2009 staging ranked from stage I to IV.

### 3.2. Outcomes

In total, 1163 patients with endometrial cancer and BMI ≥ 40 were included in the review. In total, 9 of the 12 selected studies presented intra-operative complication data, 11 reported post-operative complication rates, 9 data about severe complication rates, and 7 about complete surgical staging. The primary outcome was the intra-operative complication rate, which ranged from 0 to 3.9% for LPS, 0 to 12.5% for LPT, and 1.4 to 7.7% for the Robot approach. Secondary outcomes included the post-operative complication rate, which was, respectively, between 2.5 and 29.9% for LPS, 12.5 and 44.0% for LPT, and 8.7 and 19.6% for Robot; and the severe complication rate, which was between 0 and 25.8% in LPS, 6.9 and 25.0% in LPT, and 0 and 9.8% in Robot. Finally, complete surgical staging was achieved in the 14.5 to 37.1% range in LPS, 7.0 to 27.7% in LPT, and 20.0 to 75.0% in Robot. All those results are summarized in Table 2.

### 3.3. Meta-Analysis

The eight studies comparing the different surgical approaches were enrolled in the meta-analysis. Mini-invasive strategies (MISs) (LPS or Robot) were compared to LPT. A sub-analysis for MIS type was performed. All the results have been graphical in a dedicated Forest plot.

In total, 233 patients in the MIS arm were compared with 143 patients who underwent LPT approaches, exploring intra-operative complication outcomes.

The MIS group showed a non-significant lower intra-operative complication rate than open surgery (OR 0.68 [95% CI 0.21–2.26] *p* = 0.18) as shown in Figure 2). A sub-analysis showed an OR 0.28 [0.10–0.74] for Robot approaches and 1.18 [0.15–9.50] for LPS (*p* = 0.23) (Figure 2a).

For post-operative outcomes, 457 patients in the MIS arm were compared with 299 in the LPT, showing a non-statistically significant OR of 0.41 [0.26–0.64] (*p* = 0.27). In the type of MIS sub-analysis, the OR was 0.59 [0.09–3.73] for Robot and 0.49 [0.16–1.53] for LPS (*p* for sub-analysis = 0.88) (Figure 2b).

Lastly, no difference has been proved in the complete surgical staging for the MIS or LPT group (217 vs. 331 patients; OR 1.01 [0.42–2.45], *p* = 0.10) (Figure 2c).

### 3.4. Network Meta-Analysis

A network meta-analysis was conducted to compare all three examined approaches. The Robot approach ranked better than LPS and LPT in preventing intra-operative complications (70.7% probability of being better than LPS and 99.2% of being better than LPT) (effect size −1.04 [−1.89; −0.20 95% CI]; *p* = 0.016), with a P-score of 0.85 for Robot, 0.60 for LPS and 0.05 for LPT (Figure 3a).

Regarding post-operative complications rate, the LPS approach ranked first, with a slightly non-statistically significant *p* (50.9% probability of being better than Robot and 97.2% of being better than LPT) (effect size −0.66 [−1.34; 0.02 95% CI]; *p* = 0.056), with a P-score of 0.74 for LPS, 0.71 for Robot and 0.05 for LPT (Figure 3b).

Finally, in completing surgical staging, the Robot was statistically significantly better than the other two approaches (94.5% probability of being better than LPS and 96.2% of being better than LPT) (effect size 0.69 [−0.07; 1.44 95% CI]; *p* = 0.04), with a P-score of 0.95 for Robot, 0.33 for LPS and 0.22 for LPT (Figure 3c).

We used a node graph to demonstrate the correlation of the three approaches used in the network meta-analysis, as shown in Appendix A.

## 4. Discussion

### 4.1. Interpretation of Results

Our analysis aims to identify which surgical approach is the safest and most effective for the management of endometrial cancer in patients with BMI ≥ 40. Our meta-analysis does not show that one treatment is safer than the other in a statistically significant manner. Nevertheless, MIS approaches show a protective trend in intra-operative complications, post-operative complications, and the ability to complete surgical staging with at least bilateral sentinel lymph nodes. In contrast, the network meta-analysis that sought to make an indirect comparison between all three approaches examined showed that the robotic approach is most likely to be the safest in preventing intra-operative complications and the ability to complete lymph node staging. In contrast, it shows a superimposable performance compared to the LPS approach in preventing post-operative complications. Another side of the coin is how the severe complication rate remains very high in this patient setting, with values as high as 44% in the LPT approach. This may be due to the inherent frailty of patients with extreme BMI, where the rate of surgical site infections may be a crucial compromising factor in the hospital stay [31]. Complete surgical staging rates also remain low in all three approaches, with only one study focusing on robotic surgery reporting success rates above 50%. This proves that surgery of the pelvic lymph node compartment is complicated in this type of patient, regardless of the approach. However, the ability to rotate the instrument in robotic surgery could justify the higher performance in this approach. Moreover, obesity is not only a risk factor for surgical complications due to technical and anesthesiological difficulties but also creates a chronic proinflammatory state that increases the risk of complications, regardless of the surgical technique used. Adiposopathy is associated with elevated production of proinflammatory cytokines, such as TNF-α and IL-6, which can impair wound healing, increase the risk of infection, and worsen post-operative outcomes. Therefore, it is not surprising that minimizing surgical access may result in a lower risk of surgical site infections.

### 4.2. Comparison with Existing Literature

Severe obesity is itself proven to be a risk for intra- and post-operative complications [32]. However, our results are in line with the evidence for gastric [33] and colorectal surgery [34], where minimally invasive approaches seem to have less impact in terms of morbidity. On the other hand, a high conversion rate from minimally invasive to laparotomic approaches is reported in the literature [35], both due to the inherent difficulties associated with the surgery of extremely obese patients and to the anaesthesiological and dysventilatory difficulties that these patients may present [36]. In our analysis, the laparotomic conversion rate was not considered because it is difficult to obtain from the series in the literature. However, the incidence of intra-operative complications can indirectly measure this risk, assuming that procedures without complications account for the rate of completed MIS approaches. Regarding the higher rate of post-operative complications in laparotomic approaches, the figure is in line with what is known for abdominal surgery, where laparotomy is more readily associated with infectious and dehiscence risk [37].

### 4.3. Clinical Implication

Our study can provide a basis for planning the management of patients with severe obesity and endometrial carcinoma. The need to minimize intra- and post-operative risks is also linked to optimizing the management of health resources to reduce hospitalization days and costs for patients with complications. Although robotic surgery is generally more expensive in the short term than other techniques, reducing complications and hospital length of stay could result in cost savings in the long term, improving the efficiency of the healthcare system [38]. A significant advantage of robotic surgery may be the ability to reduce conversion rates to laparotomy. Previous studies have shown that laparoscopic surgery, while minimally invasive, may present technical difficulties that lead to forced conversions in patients with severe obesity. Due to its greater maneuverability and precision, robotic surgery reduces this need, reducing associated risks and improving post-operative outcomes.

Moreover, the awareness that the robotic approach could represent the gold standard regarding safety and efficacy should also guide administrative choices, centralizing the management of these patients in centers equipped with surgery robots. On the other hand, we cannot ignore how robotic surgery may not be accessible to all patients due to the cost or availability of equipment. Issues of equity of access may arise, especially in resource-limited settings [39].

Finally, robotic technology lends itself well to integration with other technologies. In the future, the integration of emerging technologies, such as artificial intelligence and augmented reality-assisted surgery, could further improve clinical outcomes. Artificial intelligence could be used to predict intra-operative complications based on clinical and biometric data, while augmented reality could provide real-time images of the patient’s organs and tissues, further improving surgical accuracy.

### 4.4. Strength and Limitations

To our knowledge, our study is the only one to attempt a comparison between the three different abdominal surgical approaches in patients with endometrial carcinoma and severe obesity. However, some limitations accompany the study. The first is a geographical limitation. Of the 12 studies reported, none came from outside the West, with five studies from five different Italian research groups and three from the USA. This may be related to the high obesity rate and the widespread distribution of robotic centers, mainly concentrated in these two countries [40]. Another limitation is linked to the wide range of BMI treated (40–75 kg/m^2^), which may suggest a hypothetical exponential increase in operating risks as BMI increases. Finally, a final limitation is related to the lack of detailed information regarding comorbidities. Obesity often presents associated with additional systemic issues that may also affect perioperative outcomes. However, a subgroup analysis was impossible due to the lack of information in the studies examined.

Another limitation is the lack of concrete economic considerations for the three techniques. Economic analysis of different surgical techniques is a critical element in assessing their sustainability and affordability. Robotic surgery, despite its proven clinical benefits, has significantly higher costs than laparoscopy and laparotomy. These costs include not only the initial purchase of the robotic system but also the costs associated with maintenance, consumables, and staff training. However, reduced morbidity and hospital stays play a role in reducing the economic impact of endometrial cancer in obese patients. In addition, an assessment of indirect costs, such as patients’ loss of productivity and work capacity, will be of interest in future studies. To date, we have no evidence to explore this aspect further. Finally, another economic aspect is the centralization of services. Creating centers of excellence dedicated to robotic surgery could optimize the use of resources, ensuring a higher volume of procedures and reducing the cost per procedure. However, the latter speculation risks limiting equitable access to these technologies. In low- and middle-income countries, adopting robotic surgery is often limited by economic resources. This raises ethical questions about inequitable access to quality care and underscores the need for health policies that balance technological innovation with economic sustainability [39].

## 5. Conclusions

In patients undergoing hysterectomy for endometrial carcinoma with obesity grade 3 or higher, the minimally invasive approach seems to minimize the risks of intra- and post-operative complications. In particular, the robotic approach seems to show the best strategy, also improving the lymph node surgical staging success rate. Further studies with prospective designs and direct comparisons will be necessary to confirm these data.

## Figures and Tables

**Figure 1 cancers-17-02018-f001:**
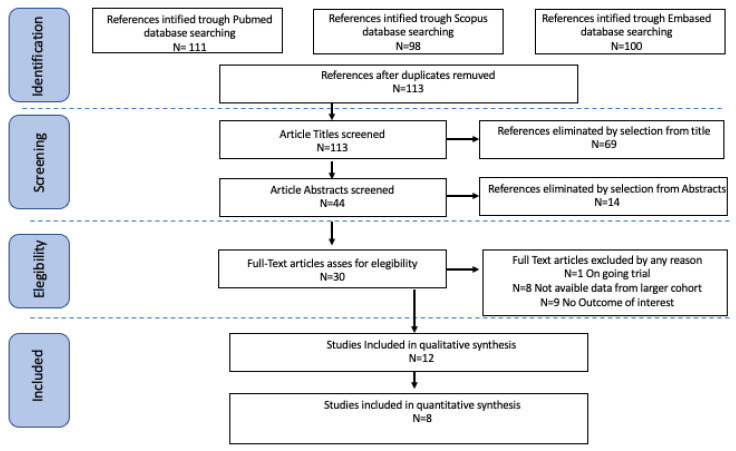
Prisma flowchart.

**Figure 2 cancers-17-02018-f002:**
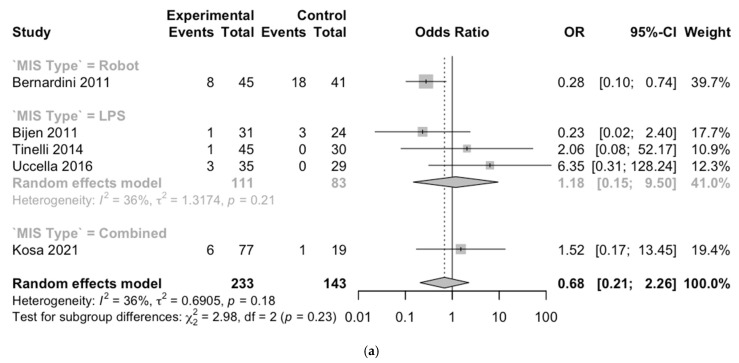
Forest plot. (**a**) Intra-operative complication. (**b**) Post-operative complication. (**c**) Complete surgical staging [23,24,25,26,27,29,30].

**Figure 3 cancers-17-02018-f003:**
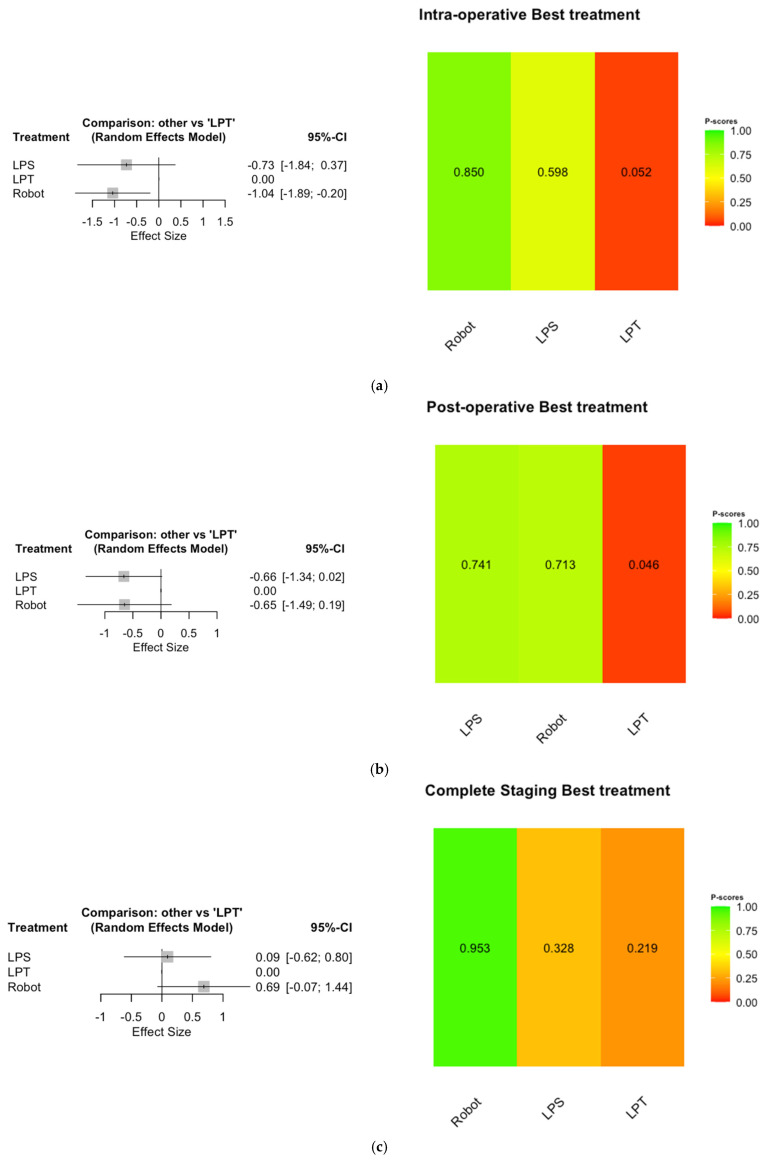
(**a**) Intra-operative complication. (**b**) Post-operative complication. (**c**) Complete surgical staging.

**Table 1 cancers-17-02018-t001:** Studies included.

Non-Comparative Studies							
Name	Country	Study Design	Study Year	Approaches	FIGO * Stage	BMI ° Range	N of Participant
Armfield 2018 [19]	International	Randomized Clinical Trial	2005–10	LPS	I–III	40–63	167 ^
Corrado 2015 [20]	Italy	Prospective Case–control Multicentric	2010–14	Robot	I–IV	40–61	70
Stephan 2015 [21]	USA	Retrospective Observational Monocentric	2005–12	Robotic	I–II	50–57	56 ^
Vizza 2019 [22]	Italy	Retrospective Case–control Multicentric	2010–18	Robotic	I–II	40–55	41 ^
Comparative Studies							
Bernardini 2011 [23]	Canada	Prospective Case–control Multicentric	2008–10	LPT vs. Robot	I–IV	40–75	86
Bijen 2011 [24]	Netherlands	Randomized Clinical Trial	2009–11	LPS vs. LPT	I–IV	40–55	55 ^
Giugale 2012 [25]	USA	Retrospective Cohort Monocentric	2001–07	LPS vs. LPT vs. Robotic	I–IV	40–61	398 ^
Kosa 2021 [26]	Canada	Prospective Observational Multicentric	2012–14	LPS vs. LPT vs. Robotic	NR	40–55	103
Mendivil 2015 [27]	USA	Retrospective Cohort Monocentric	2008–11	LPS vs. LPT vs. Robotic	I–III	40–67	53
Scambia 2018 [28]	Italy	Retrospective Case–control Multicentric	2001–17	LPS vs. Robot	I–IV	40–66	134 ^
Tinelli 2014 [29]	Italy	Retrospective Observational Multicentric	2004–13	LPS vs. LPT	I–III	35–64	75
Uccella 2016 [30]	Italy	Retrospective Observational Monocentric	2000–13	LPS vs. LPT	I–III	40–62	64 ^

^ sub-analysis of the entire cohort ° body mass index * 2009 FIGO endometrial cancer staging.

**Table 2 cancers-17-02018-t002:** Safety outcome.

Name	Treatment	Intra-Operative Complication (%)	Post-Operative Complication (%)	Severe Complication * (%)	Complete Staging ^ (%)
Armfield 2018 [19]	LPS	0.6	29.9	7.8	-
Bernardini 2011 [23]	LPT	7.3	44.0	14.6	7.0
	Robot	4.4	17.7	4.4	20.0
Bijen 2011 [24]	LPS	3.2	22.6	25.8	-
	LPT	12.5	12.5	25.0	-
Corrado 2015 [20]	Robot	1.4	8.7	7.1	38.6
Giugale 2012 [25]	LPS	-	-	-	31.0
	LPT	-	-	-	27.7
	Robot	-	-	-	34.7
Kosa 2021 [26]	LPS	0	8.3	-	-
	LPT	0	26.3	-	-
	Robot	7.7	9.2	-	-
Mendivil 2015 [27]	LPS	-	6.2	6.2	-
	LPT	-	16.7	8.3	-
	Robot	-	15.4	0	-
Scambia 2018 [28]	LPS	3.9	2.5	2.5	14.5
	Robot	3.4	8.6	8.6	31.0
Stephan 2015 [21]	Robot	-	19.6	-	75.0
Tinelli 2014 [29]	LPS	2.2	8.9	0	-
	LPT	0	30.0	10.0	-
Uccella 2016 [30]	LPS	0	14.3	8.6	37.1
	LPT	0	20.7	6.9	20.7
Vizza 2019 [22]	Robot	2.4	9.8	9.8	14.6

* Clavien–Dindo ≥ 3 ^ success rate on the attempted procedure of lymph node staging.

## Data Availability

All data and the methodological process for their calculation can be supplied under explicit request to the corresponding author and provided as an ‘.R’ file.

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
