# Peer review of "Safety and Efficacy Outcomes of Robotic, Laparoscopic, and Laparotomic Surgery in Severe Obese Endometrial Cancer Patients: A Network Meta-Analysis"

_cancers, 2025, doi:10.3390/cancers17122018_

Round 1
Reviewer 1 Report
Comments and Suggestions for Authors
Dear Authors,
Using systematic review and network meta-analysis the Authors addressed the important issue of safety of various surgical techniques in a group of patients with severe obesity and diagnosed with endometrial cancer of varying degrees of advancement . Obesity is responsible for about 40% of endometrial cancer cases . The analysis carried out showed that robotic surgery shows superior outcomes for complete lymph nodal staging in obese endometrial cancer patients, while LPS is favorable for post-operative complications.
I hope that some of the comments and remarks attached below will be useful.
Part Introduction:
I suggest replacing the first two sentences with one, e.g. as below, which will avoid several repetitions of the word severe.
Grade 3 obesity, also called severe obesity, a body mass index (BMI) greater than or equal to 40 kg/m², is a global and steadily increasing public health problem including both developed and developing countries
Part 2.4. Primary and Secondary Endpoints
Lines 118-119: The rate of complete surgical staging, defined as the bilateral removal of at least one sentinel lymph node, out of the total number of operations ? Is such a criterion for assessing the completeness of the procedure sufficient? Considering that some of the operated patients were in stage IV of advancement, the assessment of completeness should, in my opinion, also include information about any remaining persistent infiltrates. Does such a formulated assessment criterion mean that in each case of surgery from the articles selected for analysis, the surgical assessment of the lymph nodes consisted only of the sentinel node procedure???
Do the studies selected for analysis enable an assessment of the impact of cancer advancement on the safety of the analyzed surgical techniques?
Table 1: information about ^ Sub-analysis of the entire cohort requires comment: which cases were dropped from the analysis and why? Are the numbers of patients given in the table the total number or reduced by those not analyzed?
Author Response
Thank you for taking the time to work on our manuscript. We found your advice very helpful in improving the quality of our manuscript. Below are point by point corrections made in accordance with your comments.
“I suggest replacing the first two sentences with one, e.g. as below, which will avoid several repetitions of the word severe.”
We made the changes according to your example
“Lines 118-119: The rate of complete surgical staging, defined as the bilateral removal of at least one sentinel lymph node, out of the total number of operations ? Is such a criterion for assessing the completeness of the procedure sufficient? Considering that some of the operated patients were in stage IV of advancement, the assessment of completeness should, in my opinion, also include information about any remaining persistent infiltrates. Does such a formulated assessment criterion mean that in each case of surgery from the articles selected for analysis, the surgical assessment of the lymph nodes consisted only of the sentinel node procedure???”
We apologize if this definition caused confusion. The procedure was considered complete when at least one lymph node on each side was removed, when the surgical intent was the search for the sentinel lymph node. In cases of advanced pathology, the procedures addressed were lymphadenectomy, the number of lymph nodes removed being reported when reported by individual studies. We have improved this explanation in section 2.4
“Do the studies selected for analysis enable an assessment of the impact of cancer advancement on the safety of the analyzed surgical techniques?”
Unfortunately, the totality of included studies did not allow subanalysis based on stage of disease.
“Table 1: information about ^ Sub-analysis of the entire cohort requires comment: which cases were dropped from the analysis and why? Are the numbers of patients given in the table the total number or reduced by those not analyzed?”
Some included studies also analyzed patients with BMI <40, consequently, the indication that it is an extraction from a larger sample comes from taking only data from patients who met our inclusion criteria
We hope that these clarifications may have improved our work. We would also like to inform you that we have performed a general review of English and readability. In any case, an updated version of the manuscript is attached on the Journal website
Reviewer 2 Report
Comments and Suggestions for Authors
My comments:
1. On Page 2, lines 83-84, can you please specify what "MCS", "EB" and "CR" stand for? I believe these are authors on the paper. Perhaps change the sentence to "Study selection was made independently by authors MCS and EB. In case of discrepancy, author CR decided on inclusion or exclusion."
2. Page 3, lines 106-107: Please remove periods from author initials, for consistency with other formatting in manuscript.
3. Were the authors able to adjust for any extraneous factors or study year? If yes, please make this more clear in the Methods section.
4. Forest plot is spelled incorrectly in text. Please correct.
5. Manuscript should undergo thorough review for grammatical errors and misspellings.
Author Response
Thank you for taking the time to work on our manuscript. We found your advice very helpful in improving the quality of our manuscript. Below are point by point corrections made in accordance with your comments.
“On Page 2, lines 83-84, can you please specify what "MCS", "EB" and "CR" stand for? I believe these are authors on the paper. Perhaps change the sentence to "Study selection was made independently by authors MCS and EB. In case of discrepancy, author CR decided on inclusion or exclusion."
Yes, those are authors, we changed the text according to Your observation
“2. Page 3, lines 106-107: Please remove periods from author initials, for consistency with other formatting in manuscript”
We did it
“Were the authors able to adjust for any extraneous factors or study year? If yes, please make this more clear in the Methods section.“
Yes, the authors accounted for extraneous factors and study year. In the Methods section, a network meta-regression was performed to explore the potential influence of key study-level covariates, including mean BMI, menopausal status distribution, tumor stage distribution, and publication year. These covariates were included as potential effect modifiers in the analysis to reduce heterogeneity and improve the robustness of the treatment comparisons. We stated it in the new version of the Manuscript
“Forest plot is spelled incorrectly in text. Please correct.”
Thank You, we changed it into “Forest plot”
“Manuscript should undergo thorough review for grammatical errors and misspellings.”
We would also like to inform you that we have performed a general review of English and readability. In any case, an updated version of the manuscript is attached on the Journal website
Reviewer 3 Report
Comments and Suggestions for Authors
Dear authors,
I have read your interesting manuscript. The topic is interesting and usable for the readers. The methodology is good.
To improve the article, I have few suggestions:
-if possible, present the results of the follow-up of patients undergoing MIS and laparotomy
-if possible, compare the results by dividing them into stage I-II vs stage III-IV
Best Regards
Author Response
Thank you for taking the time to work on our manuscript. We found your advice very helpful in improving the quality of our manuscript. Below are point by point corrections made in accordance with your comments.
“To improve the article, I have few suggestions:
-if possible, present the results of the follow-up of patients undergoing MIS and laparotomy
-if possible, compare the results by dividing them into stage I-II vs stage III-IV”
Unfortunately, The data presented in the individual studies did not allow for any form of subanalysis. This limitation was pointed out in the Disussion. We hope that this limitation will not irreversibly compromise our manuscript.
We would also like to inform you that we have performed a general review of English and readability. In any case, an updated version of the manuscript is attached on the Journal website